# NFAT5/TonEBP Limits Pulmonary Vascular Resistance in the Hypoxic Lung by Controlling Mitochondrial Reactive Oxygen Species Generation in Arterial Smooth Muscle Cells

**DOI:** 10.3390/cells10123293

**Published:** 2021-11-24

**Authors:** Hebatullah Laban, Sophia Siegmund, Maren Zappe, Felix A. Trogisch, Jörg Heineke, Carolina De La Torre, Beate Fisslthaler, Caroline Arnold, Jonathan Lauryn, Michael Büttner, Carolin Mogler, Katsuhiro Kato, Ralf H. Adams, Hanna Kuk, Andreas Fischer, Markus Hecker, Wolfgang M. Kuebler, Thomas Korff

**Affiliations:** 1Institute of Physiology and Pathophysiology, Department of Cardiovascular Physiology, Heidelberg University, 69120 Heidelberg, Germany; h.laban@physiologie.uni-heidelberg.de (H.L.); sophia.siegmund@physiologie.uni-heidelberg.de (S.S.); maren.hoedebeck@yahoo.de (M.Z.); caro.arnold@gmx.net (C.A.); hecker@physiologie.uni-heidelberg.de (M.H.); 2Deutsches Zentrum für Herz-Kreislauf-Forschung e.V. (DZHK), Partner Site Heidelberg/Mannheim, 69120 Heidelberg, Germany; 3Department of Cardiovascular Physiology, Mannheim Medical Faculty, Heidelberg University, 69120 Heidelberg, Germany; Felix.Trogisch@medma.uni-heidelberg.de (F.A.T.); joerg.heineke@medma.uni-heidelberg.de (J.H.); 4European Center for Angioscience (ECAS), Medical Faculty Mannheim, Heidelberg University, 69120 Heidelberg, Germany; a.fischer@Dkfz-Heidelberg.de; 5NGS Core Facility, Medical Faculty Mannheim, Heidelberg University, 69120 Heidelberg, Germany; Carolina.DeLaTorre@medma.uni-heidelberg.de; 6Institute for Vascular Signalling, Goethe University, Frankfurt am Main, 60323 Frankfurt, Germany; fisslthaler@vrc.uni-frankfurt.de; 7German Center of Cardiovascular Research (DZHK), Partner site RheinMain, Frankfurt am Main, 60323 Frankfurt, Germany; 8Institute of Physiology, Charité-Universitätsmedizin Berlin, Corporate Member of Freie Universität Berlin and Humboldt Universität zu Berlin, 10099 Berlin, Germany; jonathan.lauryn@charite.de (J.L.); wolfgang.kuebler@charite.de (W.M.K.); 9Metabolomics Core Technology Platform, Centre for Organismal Studies, Heidelberg University, 69120 Heidelberg, Germany; michael.buettner@cos.uni-heidelberg.de; 10Institute of Pathology, School of Medicine, Technical University Munich, 80333 Munich, Germany; carolin.mogler@tum.de; 11Department of Tissue Morphogenesis, Faculty of Medicine, Max Planck Institute for Molecular Biomedicine, University of Münster, 48149 Münster, Germany; katsuhiro.kato@mpi-muenster.mpg.de (K.K.); ralf.adams@mpi-muenster.mpg.de (R.H.A.); 12The Ottawa Department of Medicine, Faculty of Medicine, University of Ottawa, Ottawa, ON K1N 6N5, Canada; hanna.kuk@hotmail.com; 13Division Vascular Signaling and Cancer, German Cancer Research Center (DKFZ), 69120 Heidelberg, Germany; 14Department of Internal Medicine I, Heidelberg University, 69120 Heidelberg, Germany

**Keywords:** pulmonary artery, smooth muscle cells, NFAT5, transcriptome, mitochondrial ROS

## Abstract

Chronic hypoxia increases the resistance of pulmonary arteries by stimulating their contraction and augmenting their coverage by smooth muscle cells (SMCs). While these responses require adjustment of the vascular SMC transcriptome, regulatory elements are not well defined in this context. Here, we explored the functional role of the transcription factor nuclear factor of activated T-cells 5 (NFAT5/TonEBP) in the hypoxic lung. Regulatory functions of NFAT5 were investigated in cultured artery SMCs and lungs from control (*Nfat5^fl/fl^*) and SMC-specific *Nfat5*-deficient (*Nfat5^(SMC)−/−^*) mice. Exposure to hypoxia promoted the expression of genes associated with metabolism and mitochondrial oxidative phosphorylation (OXPHOS) in *Nfat5^(SMC)−/−^* versus *Nfat5^fl/fl^* lungs. In vitro, hypoxia-exposed *Nfat5*-deficient pulmonary artery SMCs elevated the level of OXPHOS-related transcripts, mitochondrial respiration, and production of reactive oxygen species (ROS). Right ventricular functions were impaired while pulmonary right ventricular systolic pressure (RVSP) was amplified in hypoxia-exposed *Nfat5^(SMC)−/−^* versus *Nfat5^fl/fl^* mice. Scavenging of mitochondrial ROS normalized the raise in RVSP. Our findings suggest a critical role for NFAT5 as a suppressor of OXPHOS-associated gene expression, mitochondrial respiration, and ROS production in pulmonary artery SMCs that is vital to limit ROS-dependent arterial resistance in a hypoxic environment.

## 1. Introduction

Adequate adjustment of the vascular SMC tone to different environmental conditions is vital for the function of any organ as it controls the resistance of arteries and arterioles by regulating their diameter and consequently the local blood supply. In the lung, SMCs of pulmonary arteries (paSMCs) constrict or relax depending on the oxygen level to optimize gas exchange by matching alveolar perfusion to local ventilation [1]. Specifically, paSMCs can sense oxygen partial pressure and increase their tone when exposed to hypoxia—a special feature of paSMCs located in small pulmonary arteries [2] known as hypoxic pulmonary vasoconstriction (HPV). At high altitudes and under pathophysiological conditions causing alveolar hypoxia (e.g., chronic obstructive pulmonary disease (COPD) or low respiratory drive) paSMCs may be chronically forced to increase their tone. Mechanistically, an accumulating body of evidence suggests that mitochondrial function is rate-limiting for sensing as well as initiating the paSMC contraction by producing reactive oxygen species (ROS) in response to decreasing oxygen levels [1]. While HPV shows a biphasic course comprising of an acute and sustained phase, the latter and long-lasting phase was especially shown to rely on mitochondrial function [3]. In addition to HPV, chronic hypoxia also evokes structural vascular remodeling which results in the increased coverage of small pulmonary arteries and arterioles by SMCs. In mouse models of chronic hypoxia, arterial muscularization usually starts within seven days after the onset of hypoxia and originates from a preexisting SMC population that dedifferentiates, migrates, proliferates, and finally redifferentiates [4,5].

Both HPV and hypermuscularization of pulmonary arteries progressively increase pulmonary arterial resistance in chronic hypoxia. Consequently, pulmonary arterial pressure and right ventricular afterload increase, which promote right ventricular hypertrophy and functional failure. These pathophysiological mechanisms may bear fatal consequences for lung and heart function in the long run, but initially emanate from a simple adjustment of the vascular phenotype to altered environmental conditions. For instance, in endothelial cells (ECs) hypoxia activates hypoxia inducible factor 1 alpha (HIF1A), which promotes the expression of *Pdgfb* (platelet-derived growth factor subunit B) [6]. In turn, PDGFB stimulates the activity of the Krüppel-like (transcription) factor 4 (KLF4) in paSMCs as a prerequisite for their expansion and migration [6]. As such, the transcriptional adjustment precedes the functional adaptation of paSMCs coping with hypoxic stress.

In this context, we aimed to elucidate the role of the nuclear factor of activated T-cells 5 (NFAT5)—a stress response transcription factor. NFAT5 or tonicity enhancer binding protein (TonEBP) is a calcineurin-independent member of the Rel family of transcription factors and has recently been described as an immunometabolic stress protein that modifies the cellular transcriptome in a plethora of pathophysiological conditions including hyperlipidemia, insulin resistance, arteriosclerosis, rheumatoid arthritis, hypertonicity [7,8], and ischemia [9]. In vascular SMCs, NFAT5 controls different aspects of their phenotypic modulation including the angiotensin II-mediated upregulation of the SMC marker α-smooth muscle cell actin (αSMA) [10]. Recently, we revealed that NFAT5 also controls the adaptive responses of vascular SMCs exposed to biomechanical stress and hypertension by regulating the expression of genes functionally attributed to proliferation, matrix synthesis, and cytoskeletal reorganization [11,12,13,14]. The relevance of NFAT5 for the response of vascular SMCs to hypoxia, however, has not been studied so far. To this end, we utilized a mouse model to ablate *Nfat5* specifically in SMCs [14,15,16] and investigated the functional impact of NFAT5 on pulmonary artery SMCs during exposure to chronic hypoxia in vitro and in vivo.

## 2. Materials and Methods

### 2.1. Cell Culture

Mice were euthanized by carbon dioxide inhalation and subsequent cervical dislocation. Murine aortic smooth muscle cells (aoSMCs) were isolated from the aorta of *Nfat5^fl/fl^* mice as described earlier [14]. Briefly, after carefully removing the adventitia, the aorta was washed twice in DPBS w/o calcium and magnesium, cut in 1 mm-sized rings, and digested with 1% collagenase overnight. All vascular smooth muscle cells were cultured in DMEM low glucose medium (ThermoFisher/Gibco, Karlsruhe, Germany) containing 15% FCS.

Murine pulmonary artery smooth muscle cells (mpaSMCs) were isolated from pulmonary arteries as described by Lee et al. [17]. In brief, a slurry of agarose and iron particles in Panserin (PAN-Biotech, Aidenbach, Germany) was infused into the pulmonary circulation via the right ventricle of the heart (RV). The particles cannot pass the capillary system and remain in arteries and arterioles. Upon solidification of the agarose, artery segments were magnetically extracted from dissociated lung tissue. Cells were isolated by collagenase treatment, suspended, and cultured as described for aoSMCs. Cell identity was routinely tested by performing immunofluorescence-based or protein biochemical methods to detect SMC markers (Appendix A). Cultured cells were exposed to hypoxia for up to 24 h by transferring them to a hypoxia chamber (Invivo2 Plus, Ruskinn, Bridgend, UK) that was pre-equilibrated with 1% O_2_.

### 2.2. Viral Transduction

Deletion of the floxed *Nfat5* exon was achieved by transduction with an adenovirus expressing Cre recombinase (iCre) under a cytomegalovirus (CMV) promoter (Ad-CMV-iCre, Cat. no. 1045, Vector Biolabs, Malvern, PA, USA). The virus was applied to the cells with a multiplicity of infection (MOI) of 500. An adenovirus with an empty CMV promoter (Ad-CMV-Null, Cat. no. 1300, Vector Biolabs) served as a control. Cells were utilized for experiments 48 h after exposure to adenoviral vectors. The knockout efficiency was controlled by determining the protein level of NFAT5 in cell lysates (Appendix A).

### 2.3. Immunofluorescence Analyses of Cultured Cells

Cells were fixed in ice-cold methanol for 15 min and allowed to dry for 20 min. Rehydrated cells were blocked for 30 min with 5% donkey serum in TRIS-buffered salt solution (TBS) supplemented with 0.05% Tween 20. Cells were incubated for 2 h with the mouse anti-NFAT5 (1:100, direct label, NFAT5—Alexa 546, sc-398171), washed with TBS, and mounted with Mowiol (Merck/Calbiochem, Darmstadr, Germany). Nuclei were visualized by DAPI. Fluorescence images were recorded by using an Olympus IX83-System (Olympus, Hamburg, Germany) in combination with the cellSens Software (v. 1.12).

### 2.4. Analysis of Metabolic Activity in Cultured Cells

Glycolysis and mitochondrial respiration of mpaSMCs were automatically assessed by real-time analysis of the oxygen consumption rate (OCR, determines oxygen consumption/min) and extracellular acidification rate (ECAR, determines changes in pH) over a period of 90 min using a Seahorse XF analyzer (Agilent, Waldbronn, Germany). The mpaSMCs were treated with adeno-associated virus to overexpress CRE recombinase (*Nfat5^−/−^*) or GFP (*Nfat5^fl/fl^*) and exposed to normoxia/hypoxia for 24 h prior to the analysis following manufacturer’s instructions (protocol I: Seahorse XF Glycolysis Stress Test, protocol II: Seahorse XF Cell Mito Stress Test).

OCR was determined while treating the cells with oligomycin (OM, mitoch. complex V/ATP synthase inhibitor), FCCP (a protonophore to uncouple mitochondrial respiration), and rotenone/antimycin A (Rot/Antim., complex I/III inhibitors blocking the electron transport chain) to analyze ATP-linked respiration, proton leak respiration, and mitochondrial (maximal) respiratory capacity. ECAR was determined while treating the cells with glucose (Gluc.), oligomycin (OM, blocks mitochondrial ATP generation to determine the max. glycolytic capacity), and 2-deoxy-d-glucose (2-DG, glucose analog to block glycolysis) to analyze basal glycolysis, glycolytic reserve, maximal glycolysis, and nonglycolytic ECAR. Both OCR and ECAR were normalized to the total cellular protein per sample.

### 2.5. Detection of Radical Oxygen Species (ROS) in Cultured Cells

Cells were exposed to normoxia or hypoxia for 24 h. Mitochondrial ROS production was traced by supplementing the medium with MitoTracker™ Red CM-H2Xros (Cat. no. M7513, ThermoFisher, Karlsruhe, Germany) for 30 min according to the manufacturer’s instructions. This probe is oxidized by reactive oxygen species to become a mitochondrion-selective fluorophore. Cells were washed with HBSS without phenol red and immediately analyzed by recording the fluorescence in randomly selected areas of the culture surface utilizing a fluorescence microscope (IX83, Olympus, Hamburg, Germany). Fluorescence images were automatically analyzed by applying the count & measure module of the cellSens software (Olympus, Hamburg, Germany, version 1.18) that allows detection and gray value assessment of the mean mitochondria-associated fluorescence signal within a region of interest (Appendix A). MitoTracker™ Green FM (Cat. no. M7514, ThermoFisher, Karlsruhe, Germany) was utilized to assess the total cellular mitochondrial density (Appendix A). MitoTracker™ Red CM-H2Xros–based results were controlled by MitoSOX Red (Cat. no. M36008, ThermoFisher, Karlsruhe, Germany), another mitochondria-specific ROS-sensitive reagent that is oxidized by superoxide and becomes fluorescent upon binding to nucleic acids (Appendix A).

### 2.6. Analysis of Gene Expression

RNA was extracted by lysing cells or lung tissue in RLT buffer (Cat. no. 79216, Qiagen, Hilden, Germany) containing 1% β-mercaptoethanol and subsequently purified using the RNeasy Mini Kit (Cat. no. 74106, Qiagen, Hilden, Germany) and RNase-Free DNase Set (Cat. no. 79254, Qiagen, Hilden, Germany) according to the manufacturer’s instructions. RNA quantity and quality were examined (NanoDrop ND-1000 Spectrophotometer, NanoDrop Technologies, Wilmington, DE, USA). Depending on RNA quantity, cDNA was synthesized utilizing the Sensiscript Reverse Transcription Kit (Cat. no. 205213, Qiagen, Hilden, Germany) or the Omniscript Reverse Transcription Kit (Cat. no. 205113, Qiagen, Hilden, Germany) and Oligo(dT)15 Primer (C110A, Promega, Madison, WI, USA). Quantitative real-time RT-PCR was performed in a Rotor-Gene Q (Qiagen, Hilden, Germany) using the 5x QPCR Mix EvaGreen (No Rox) (BS76.590.5000, Bio&SELL GmbH, Feucht, Germany) and water for Molecular Biology (Cat. no. H20MB0106, Merck Millipore, Darmstadt, Germany). PCR was conducted in 40 cycles of denaturation (95 °C, 15 s), annealing (temperature depending on primer, 25 s), and elongation (72 °C, 10 s) after one initial phase of 95 °C for 15 min to activate the DNA polymerase. Fluorescence was examined at the end of the annealing phase (excitation at 470 nm, emission at 530 nm). The threshold cycle (Ct) was set within the exponential phase of the PCR using the Rotor-Gene Q Series Software (v.2.3.1, Build 49, Qiagen, Hilden, Germany). The PCR product was quantified using the ΔΔCt method. Amplification of the ribosomal protein S12 (Rps12) served as an internal standard. The following mouse primers were utilized: *Ptgs2-for 5′-TGAGCAACTATTCCAAACCAGC-3′/rev 5′-GCACGTAGTCTTCGATCACTATC-3′, Nfat5-for 5′-AACATTGGACAGCCAAAAGG-3′/rev 5′-GCAACACCACTGGTTCATTA-3′, Rpl32-for 5′-GGGAGCAACAAGAAAACCAA-3′/rev 5′-ATTGTGGACCAGGAACTT GC-3′, Rps12-for 5′-GAAGCTGCCAAAGCCTTAGA-3′/rev 5′-AACTGCAACCAACCACCTTC-3′, Actb-for 5′-CGGTTCCGATGCCCTGAGGCTCTT-3′/rev 5′-CGTCACACTTCATGATGGAATTGA-3′, Vegfa-for 5′-GTACCTCCACCATGCCAAGT-3′/rev 5′-ACTCAAGGGCTTCATCGTTA-3′.*

For PCR-based expression profiling, total RNA was isolated from cultured cells as described. cDNA was synthesized utilizing the RT2 First Strand Kit (Cat. no. 330401, Qiagen, Hilden, Germany). The RT^2^ Profiler PCR arrays “Mouse Mitochondrial Energy Metabolism” (Cat. No. 330231 PAMM-008ZR, Qiagen, Hilden, Germany) were performed according to manufacturer’s instructions using the RT2 SYBR Green ROX FAST Mastermix (Cat. no. 330620, Qiagen, Hilden, Germany). Real-time PCR was run in a Rotor-Gene Q (Qiagen, Hilden, Germany) following the cycling conditions of the manufacturer. Threshold cycle (Ct) calculation and subsequent data analysis were accomplished as described in the RT2 Profiler PCR Array Handbook (06/2019, Qiagen, Hilden, Germany). Beta-actin (*Actb*) and Beta-2-Microglobulin (*B2m*) were chosen as reference genes for analysis under the chosen experimental conditions.

### 2.7. Genome Array-Based Analyses

Gene expression profiling was performed using the GeneChip^®^ Mouse Gene 2.0 ST Array (902118, ThermoFisher/Affymetrix, Karlsruhe, Germany). A custom CDF Version 22 with ENTREZ-based gene definitions was used to annotate the arrays [18]. The raw fluorescence intensity values were normalized by applying quantile normalization. Pathways belonging to various cell functions such as cell cycle or apoptosis were obtained from public external databases (KEGG, http://www.genome.jp/kegg, accessed on 15 November 2021). A commercial software package SAS JMP15 Genomics, version 10, from SAS (SAS Institute, Cary, NC, USA) was used to analyze differential gene expression based on one way-ANOVA. A false positive rate of a = 0.05 with false discovery rate correction was taken as the level of significance. The false discovery rate (FDR) value is the adjusted *p*-value, which estimates the probability of obtaining the observed sample results when the null hypothesis is actually true. The probability value ranges between 0 (0%) and 1 (100%). A false discovery rate of 1 means 100% (i.e., 100% probability of obtaining the observed sample results when the null hypothesis [tested with ANOVA] is actually true). Gene Set Enrichment Analysis (GSEA) was applied to determine if defined lists (or sets) of genes exhibit a statistically significant bias in their distribution within a ranked gene list using the software R v3.4.0 (R Core Team 2017) and RStudio: Integrated development environment for R (RStudio Boston, MA, USA) and the fgsea package (Sergushichev, 2016).

The raw microarray data underlying this article is available in the Gene Expression Omnibus (GEO) database as record GSE178468 at https://www.ncbi.nlm.nih.gov/geo/query/acc.cgi?acc=GSE178468, accessed on 15 November 2021).

### 2.8. Mouse Model

All animal studies were performed with permission of the Regional Council Karlsruhe (permission number: 35-9185.81/G-233/18) and conformed to the Guide for the Care and Use of Laboratory Animals published by the US National Institutes of Health (NIH Publication No. 85-23, revised 1996) as well as Directive 2010/63/EU of the European Parliament on the protection of animals used for scientific purposes. *Nfat5^fl/fl^* mice were described earlier [15] and crossed with SM-MHC-CreER^T2^ [16] (only male mice carry the construct) kindly provided by Stefan Offermanns and Nina Wettschureck (Max Planck Institute for Heart and Lung Research, Bad Nauheim, Germany). All mice were housed following GV-SOLAS recommendations and were hygienically monitored by sentinel animals. The genetic ablation of NFAT5 in the corresponding offspring was induced in male mice (age: 10–12 weeks) by application of 1 mg tamoxifen per day i.p. for 5 consecutive days (*Nfat5^(SMC)−/−^*) or miglyol as solvent control (*Nfat5^fl/fl^*). Mice were used in experiments after a recovery period of 2–3 weeks.

Mice were exposed to hypoxia for 7 or 21 days by placing them into a hypoxia chamber (A-Chamber/ProOx O_2_ controller, Biospherix, Parish, NY, USA) with free access to drinking water and fodder (humidity: 50–65%, temperature: 21–23 °C) with 10% oxygen and 90% nitrogen.

### 2.9. Capillary Electrophoresis and Protein Analyses

Whole cell lysates from murine vascular SMCs or from frozen lung lobes were prepared in a radioimmunoprecipitation assay (RIPA) buffer. Lysates from cytosolic or nuclear fractions were generated by lysing cells in buffer I containing 10 mM HEPES, 10 mM KCl, 1 µM EDTA, 1 µM EGTA, 15% Nonidet, protease, and phosphatase inhibitors. After centrifugation (12,000× *g* at 4 °C for 15 min) the supernatant containing the cytosolic protein fraction was processed for further analyses. The remaining sediment was dissolved in 40 µL buffer II consisting of 20 mM HEPES, 400 mM NaCl, 0.01 M EDTA, 0.01 M EGTA, 15% Nonidet, and protease and phosphatase inhibitors. Subsequently, this solution was sonicated (ultrasound) two times for 5 s at 50 Watts at 4 °C. After centrifugation (12,000× *g* at 4 °C for 15 min) the supernatant containing the nuclear protein fraction was processed for further analyses.

Lysates were processed for capillary electrophoresis utilizing the Simple Western technology (WES, ProteinSimple, Bio-Techne, San Jose, CA, USA). Sample preparation and antibody application were performed according to the manufacturer’s instructions using the 12–230 kDa Wes Separation Module (Cat. no. SM-W004, ProteinSimple, Bio-Techne, San Jose, CA, USA). The following antibodies were applied (Cat. no., supplier, dilution): anti-NFAT5 (NB120-3446, Bio-Techne/Novus Biologicals, Wiesbaden, Germany, 1:10), anti-αSMA (ab5694, Abcam, Cambridge, UK, 1:10), anti-αtubulin (CS #2144, Cell Signaling, Danvers, MA, USA, 1:10), anti-HDAC1 (NB100-56340, Bio-Techne/Novus Biologicals, 1:10), anti-β-actin (MAB8929, Bio-Techne/R&D Systems, Wiesbaden, Germany, 1:500), anti-SMMHC (ebio 14-6400-80, Invitrogen, Waltham, MA, USA, 1:10), anti-calponin (ab46794, Abcam, 1:50), and anti-VCP (ab11433, Abcam, 1:50). Valosin containing protein (VCP) or beta-actin (β-actin) served as the loading reference. Protein abundance in the cytosolic and nuclear fraction was normalized to α-tubulin and HDAC1, respectively. Protein levels were determined by utilizing the Compass software (version 4.0.1, Build 0812, ProteinSimple, Bio-Techne, Wiesbaden, Germany).

### 2.10. Metabolomics

Mice were euthanized by cervical dislocation right after removing them from the hypoxia chamber. Metabolites from frozen lung lobes were extracted in 360 µL of 100% MeOH for 15 min at 70 °C with vigorous shaking. As internal standards, 20 µL ribitol (0.2 mg/mL) and 10 µL heptadecanoic acid (0.2 mg/mL) were added to each sample. After the addition of 200 µL, chloroform samples were shaken at 37 °C for 5 min. To separate polar and organic phases, 400 µL of water was added and samples were centrifuged for 10 min at 11,000× *g*. For derivatization, 700 µL of the polar (upper) phase was transferred to a fresh tube and dried in a speed-vac (vacuum concentrator) without heating. Derivatization (methoximation and silylation) was performed by re-dissolving the aqueous phase in 20 µL of methoximation reagent containing 20 mg/mL methoxyamine hydrochloride (Merck/Sigma, Darmstadt, Germany: 226904) in pyridine (Sigma 270970) and incubated for 2 h at 37 °C with shaking. For silylation, 32.2 µL N-Methyl-N-(trimethylsilyl)trifluoroacetamide (MSTFA; Sigma M7891) and 2.8 µL alkane standard mixture (50 mg/mL C_10_–C_40_; Fluka 68281) were added to each sample. After incubation for 30 min at 37 °C, samples were transferred to glass vials for gas chromatography/mass spectrometry (GC/MS) analysis. A GC/MS-QP2010 Plus (Shimadzu^®^, Kyoto, Japan) fitted with a Zebron ZB 5MS column (Phenomenex^®^, Hong Kong; 30 m × 0.25 mm × 0.25 µm) was used for GC/MS analysis. The GC was operated with an injection temperature of 230 °C and a 1 µL sample was injected with split mode (1:10). The GC temperature program started with a 1 min hold at 40 °C followed by a 6 °C/min ramp to 210 °C, a 20 °C/min ramp to 330 °C, and a bake-out for 5 min at 330 °C using helium as the carrier gas with constant linear velocity. The MS was operated with an ion source and interface temperatures of 250 °C, a solvent cut time of 7 min, and a scan range (m/z) of 40–700 with an event time of 0.2 s. The “GCMS solution” software (Shimadzu^®^) was used for data processing.

### 2.11. Determination of the Right Ventricular Systolic Pressure (RVSP)

For assessment of RVSP, mice were anesthetized by isoflurane (concentration: 2%) until surgical tolerance was reached (toe reflex test). A pressure-volume catheter (PVR-1030, Millar) was surgically inserted and advanced into the right ventricle through the external jugular vein. The overall blood oxygenation was monitored by a pulse oximeter (MouseSTAT, Kent Scientific, Torrington, CT, USA) throughout the procedure. Pressure pulses were continuously recorded (LabChart software, v8.1.19, ADI Instruments, Durham, NC, USA) for several minutes once a characteristic RV pressure profile was observed. RVSP values were determined by calculating the mean of at least ten pressure peaks. The influence of the oxygen concentration on the RVSP was assessed by applying different isoflurane/gas mixtures with 99%, 21%, and 10% oxygen during the RVSP measurement. For one experimental approach, physiological sodium chloride solution (solvent control) or MitoTEMPO (10 mg/kg, #SML0737-5MG, Merck/Sigma, Darmstadt, Germany) was applied by intraperitoneal injection (volume: 100 µL) 5–10 min before starting the RVSP measurement.

### 2.12. Generation and Analysis of Vibratome Sections

Mice were euthanized and perfused with 6% gelatine/PBS (37 °C) via the right ventricle. Subsequently, lungs were inflated by filling the lung with 1% (low melt) agarose/PBS (37 °C) via the trachea. After solidification, lung lobes were removed and fixed in 2% formaldehyde/PBS (4 °C) for 18 to 24 h. Next, 150 µm thick sections of lung lobes were prepared by vibratome sectioning, rinsed in PBS, and blocked for 60 min in PBS containing 5% donkey serum/0.5% Triton X100 (room temperature). Subsequently, primary antibodies in PBS/5% donkey serum/0.5% Triton X100 (mouse anti-NFAT5 (direct label Alexa 546), sc-398171, Santa Cruz Biotechnology, Heidelberg, Germany, dilution: 1:100; goat anti-CD31, AF3628, biotechne, dilution 1:200; anti αSMA-Cy3, C6198 Sigma, dilution 1:700) were added for 24 h (4 °C). After rinsing in PBS, sections were incubated with the secondary antibody (anti-goat-AlexaFluor488, 705-546-147, Dianova, Hamburg, Germany, 1:200) for 24 h (4 °C), rinsed and counterstained with DAPI and mounted with Mowiol. Fluorescence was imaged by using an Olympus IX81 confocal microscope (Olympus) and 3D image stacks were generated of regions of interest containing complete segments of arteries and arterioles. SMC coverage and blood vessel diameter of at least ten image stacks per lung were assessed by applying the morphometric tools of the Olympus Xcellence software (version 2.0, Build 4768).

### 2.13. Echocardiography

Transthoracic echocardiography was performed on anesthetized mice (1–3% isoflurane and 1 Lpm oxygenated room air) by using a Vevo 2100 high-resolution system (Visualsonics, Toronto, ON, Canada) and a 40-MHz MS-550D transducer. Two-dimensional B-mode tracings were recorded in both parasternal long and short axis views at the level of the papillary muscles and the pulmonary artery (PA), respectively, followed by one-dimensional M-mode tracings in both axes at the papillary level or pulsed-wave (PW) Doppler measurement of the peak flow in the PA. The right ventricle was recorded in B- and M-mode in a modified parasternal long axis view with an adjusted angle focussing the RV. Data were analysed offline using VevoLab 3.2.6 and the integrated cardiac measurement package. Ventricular wall thickness at end-diastole was used to characterize RV and LV microanatomy while the change of right- or left-ventricular diameter length, respectively, from end-diastole to end-systole was used to judge contractility and calculate RV or LV fractional shortening (FS). Pulmonary artery hypertension was correlated as previously described utilizing the ratio of pulmonary artery acceleration over ejection time in the PW diagram [19]. Cardiac output was monitored using the integral of PA flow, PA diameter, and heart rate. Three consecutive cardiac cycles were used for every analysis.

### 2.14. Statistical Analysis

Statistical analyses were performed by applying GraphPad Prism (Ver 9.2, GraphPad Software, San Diego, CA, USA). If not stated otherwise, all results are expressed as mean ± SD. Outliers were identified by application of the Grubbs’ test with α set to 0.05. Differences among normally distributed values of two experimental groups were analyzed by unpaired Student’s *t*-test or one sample *t*-test if applicable. *p* < 0.05 was defined as statistically significant. Differences in one parameter between normally distributed values of three or more experimental groups were analyzed by one-way ANOVA followed by Šídák’s multiple comparisons test. *p* < 0.05 was considered statistically significant (* *p* < 0.05, ** *p* < 0.01, *** *p* < 0.001).

## 3. Results

### 3.1. Hypoxia Triggers Nuclear Translocation of NFAT5 in Murine Vascular SMCs

Considering the relevance of NFAT5 for preserving cellular functions under environmental stress, we assumed that hypoxia may also trigger its expression and activity. In cultured murine vascular SMCs exposed to hypoxic conditions, *Nfat5* expression was stimulated within 24 h (Figure 1A). Nuclear translocation of NFAT5 is a prerequisite for its transcriptional activity and was assessed by applying capillary electrophoresis-based analyses of cytoplasmic and nuclear extracts from normoxia/hypoxia-exposed SMCs. Comparable with earlier analyses of SMCs exposed to biomechanical stress [12], hypoxia triggered nuclear accumulation of NFAT5 within 24 h (Figure 1B,C). The activity of NFAT5 was indirectly evidenced by exemplarily analyzing the expression of its transcriptional target *Ptgs2* [14,20]. Additionally, *Vegfa* expression was analyzed as a well-established hypoxia-triggered response that is controlled by HIF1α and was assumed to be independent of NFAT5. While expression of *Nfat5*, *Vegfa*, and *Ptgs2* was increased in SMCs upon exposure to hypoxia for 24 h, knockout of *Nfat5* inhibited *Ptgs2* but not *Vegfa* expression (Figure 1D,E, Appendix A). Corresponding analyses of murine pulmonary artery SMCs isolated from *Nfat5^fl/fl^* lungs (mpaSMCs, Appendix A) revealed comparable results (Figure 2). Considering these data, NFAT5 activity may influence early transcriptional responses of hypoxia-exposed vascular SMCs.

### 3.2. SMC-Specific Ablation of Nfat5 Stimulates the Expression of Genes Associated with Cellular Metabolism in the Hypoxic Lung

Hypoxia-sensitive responses of vascular SMCs are crucial for the proper physiological functioning of the lungs. Alveolar hypoxia induces constriction of pulmonary artery SMCs (paSMCs) to increase pulmonary vascular resistance while adjusting distribution of the blood for optimized oxygenation [21]. However, chronic generalized hypoxia additionally elevates pulmonary artery resistance by driving the muscularization of distal pulmonary arterioles thereby raising the afterload of the right ventricle. In this scenario, we studied the relevance of NFAT5 for the function of SMCs by applying a mouse model that allows for inducible and SMC-specific knockout of *Nfat5* (*Nfat5^(SMC)−/−^*) as introduced earlier [14,22].

Exposure to hypoxia for 3 to 7 days elevated the protein level of NFAT5 in paSMCs of control (*Nfat5^fl/fl^*) mouse lungs, which declined in *Nfat5^(SMC)−/−^* mice (Appendix A). Analyses of tissue lysates indicated a slight increase in NFAT5 protein abundance in hypoxic (7 d) *Nfat5^fl/fl^* but not *Nfat5^(SMC)−/−^* lungs (Figure 3A). Exposure of mice to hypoxia for 21 days did not much change the overall NFAT5 protein abundance (Figure 3A) that also showed no difference when comparing *Nfat5^fl/fl^* and *Nfat5^(SMC)−/−^* lungs.

Based on these findings, we next analyzed changes in gene expression evoked by genetic ablation of NFAT5 in SMCs. To this end, whole genome microarray analyses were performed on RNA prepared from *Nfat5^fl/fl^* lungs after exposure to normoxia and hypoxia for 7 days (Appendix A)—a time point at which paSMCs actively cover distal arterioles in this mouse model [5]. Subsequent gene set enrichment analyses (GSEA) showed augmented expression of genes associated with extracellular matrix remodeling (ECM-receptor interaction) and cell proliferation (cell cycle, DNA replication) while metabolism-associated gene expression was downregulated at large (Figure 3B).

Additionally, we compared the transcriptome of lungs from *Nfat5^(SMC)−/−^* and *Nfat5^fl/fl^* mice, which had both been exposed to hypoxia for 7 days (Appendix A). While none of the experimental conditions induced significant changes in inflammation-associated and HIF1α- as well as NFκB-controlled gene expression at the investigated time point, SMC-specific knockout of *Nfat5* led to a decline in transcripts encoding proteins regulating SMC contraction (Figure 3B). In line with the reported role of NFAT5 as a suppressor of gene expression related to thermogenesis [23] and peroxisome proliferator-activated receptors (PPAR) in adipocytes [24], transcript levels of corresponding gene sets were elevated (Figure 3B). Notably, metabolism- and especially OXPHOS-associated gene expression was increased at large in hypoxia-exposed lungs of *Nfat5^(SMC)−/−^* as compared to *Nfat5^fl/fl^* mice, which was corroborated by the accumulation of multiple partially tricarboxylic acid (TCA) cycle-associated metabolites as evidenced by gas chromatography-mass spectrometry-based analyses (Figure 3C).

Further expression analyses of individual genes in *Nfat5^fl/fl^* lungs revealed elevated levels of *Pdgfb* and SMC markers *Acta2* and *Tagln* in hypoxic versus normoxic *Nfat5^fl/fl^* lungs (Appendix A), which may be indicative of enhanced SMC coverage of smaller arteries [4]. None of these transcripts was affected by ablating *Nfat5* (Appendix A) suggesting that the level of arterial muscularization was unaltered in hypoxia-exposed *Nfat5^(SMC)−/−^* versus *Nfat5^fl/fl^* mice. In fact, a detailed histological assessment of the pulmonary arteriolar SMC coverage displayed an increase in hypoxia-mediated SMC proliferation and coverage but did not reveal any significant difference when comparing *Nfat5^(SMC)−/−^* and *Nfat5^fl/fl^* mice (Figure 4, Appendix A). However, expression of genes encoding proteins that directly or indirectly influence SMC contraction (e.g., endothelin receptor type A (*Ednra*), phospholipase C beta 1 (*Plcb1*)) were downregulated in hypoxia-exposed *Nfat5^(SMC)−/−^* lungs (Appendix A).

Additional GSE analyses of whole genome microarray data obtained at a later time point (21 days hypoxia) showed only marginal alterations in gene expression when comparing *Nfat5^(SMC)−/−^* and *Nfat5^fl/fl^* lungs (data not shown), suggesting limited activity of NFAT5 in paSMCs during later stages of chronic hypoxia.

### 3.3. Loss of Nfat5 in SMCs Stimulates the Expression of OXPHOS-Associated Genes and Accumulation of Metabolites in the Lung

We ranked the rise in OXPHOS-associated gene expression in hypoxia-exposed *Nfat5^(SMC)−/−^* lungs and the dysregulation of metabolites as the most intriguing finding. As all of these observations pointed towards a mitochondrial dysfunction, we were specifically interested in the upregulation of genes involved in the control of mitochondrial OXPHOS, which are usually downregulated under hypoxia in *Nfat5^fl/fl^* lungs (Figure 3B). To assess whether the observed transcriptomic changes may in fact originate from pulmonary artery SMCs, we investigated whether genetic ablation of *Nfat5* in isolated mpaSMCs alters the expression of genes related to mitochondrial OXPHOS functions. Subsequent expression profiling of cultured hypoxia-exposed mpaSMCs showed that loss of *Nfat5* elevates the level of transcripts encoding subunits of mitochondrial complex I (NADH dehydrogenase), complex II (succinate dehydrogenase), complex III (Ubiquinol-cytochrome C oxidoreductase), complex IV (cytochrome C oxidase), and complex V (ATP synthase) (Figure 5, left panel), which were for the most part also upregulated in the hypoxic *Nfat5^(SMC)−/−^* mouse lung (Figure 5, right panel).

### 3.4. Nfat5-Deficient mpaSMCs Promote Mitochondrial Oxygen Consumption and ROS Production under Hypoxic Conditions

Considering the expression pattern of metabolism- and mitochondrion-associated genes in hypoxic *Nfat5*-deficient mpaSMCs, we assumed that mitochondrial respiration is altered in these cells. To indirectly assess this parameter, we applied Seahorse-based analyses allowing automated measurement of the oxygen consumption rate (OCR) in real-time. Individual functional features of mitochondria (e.g., ATP synthase activity) were disabled by specific inhibitors (see Material and Methods for details) to allow the determination of baseline respiration, respiratory capacity, and ATP-linked respiration. Corresponding parameters were significantly increased in hypoxia- but not normoxia-exposed *Nfat5^−/−^* versus *Nfat5^fl/fl^* mpaSMCs (Figure 6A,B). In contrast, the extracellular acidification rate (ECAR)—a parameter assessing cellular glycolysis—decreased slightly but not significantly under these conditions (Appendix A).

Mitochondrial respiration is discussed as a major source of reactive oxygen species (ROS) in hypoxic cells [25] that is involved in the control of the sustained phase of hypoxic vasoconstriction [3]. Consequently, we investigated whether the observed amplification of mitochondrial respiration may lead to an increase in ROS production. Genetic ablation of *Nfat5* amplified the accumulation of ROS-sensitive fluorophores in the mitochondria of hypoxia-exposed mpaSMCs (Figure 6C, Appendix A). This effect was inhibited by the mitochondrion-targeting ROS scavenger MitoTEMPO [26] (Appendix A).

### 3.5. Right Ventricular Systolic Pressure (RVSP) Is Elevated in Hypoxia-Exposed Nfat5^(SMC)−/−^ Mice 

To investigate the impact of the phenotype of *Nfat5*-deficient mpaSMCs on the resistance of pulmonary arteries, we further assessed the right ventricular systolic pressure (RVSP; mean values determined for C57BL6/J (wild type) mice: ~22 mm Hg [27]) as a parameter that correlates to the resistance of pulmonary arteries. To closely mimic normoxic and hypoxic environmental conditions while assessing RVSP values, mice were exposed to 21% O_2_ and 10% O_2_ while being anesthetized_._ We initially investigated whether *Nfat5^fl/fl^* and *Nfat5^(SMC)−/−^* mice show a spontaneous increase in RVSP in response to declining (21% → 10% O_2_) oxygen levels (acute HPV) but did not observe significant differences (Figure 7A,B). Subsequently, RVSP values were determined after exposing *Nfat5^fl/fl^* and *Nfat5^(SMC)−/−^* mice to normoxia (21% O_2_) and hypoxia (10% O_2_) for 7 days. While all hypoxia-exposed mice showed elevated RVSP levels as compared to normoxia, SMC-specific loss of *Nfat5* significantly amplified the hypoxia-induced increase in RVSP (Figure 7C,D). This effect was observed under normoxic and hypoxic measurement conditions suggesting that it was independent of acute HPV responses to declining oxygen levels.

We assumed amplified mitochondrial ROS production in *Nfat5^(SMC)−/−^* pulmonary arteries as the primary cause of the observed increase in RVSP. Consequently, we tested whether mitochondrion-specific scavenging of ROS prevents out-of-proportion pulmonary hypertension in hypoxia-exposed *Nfat5^(SMC)−/−^* mice. To this end, hypoxia-exposed mice were treated with MitoTEMPO that normalized the RVSP values in *Nfat5^(SMC)−/−^* versus *Nfat5^fl/fl^* mice (Figure 7E).

### 3.6. Loss of NFAT5 in SMCs Impairs Right Heart Function in Hypoxia-Exposed Mice

We finally investigated whether the raise in RVSP in *Nfat5^(SMC)−/−^* mice after exposure to hypoxia for 7 days may influence right ventricular functions and morphometry in the long run. Thus, corresponding parameters were assessed after exposing mice to hypoxia for 21 days. Subsequent echocardiographic analyses revealed an unchanged output of the right ventricle (Figure 8A) but the pulmonary artery acceleration/ejection time (PAT/ET) ratio, as well as right ventricular fractional shortening (correlating with ventricular function), were significantly decreased in *Nfat5^(SMC)−/−^* mice after exposure to hypoxia (Figure 8B,C, Appendix A). Moreover, the thickness of the right ventricular wall and interventricular septum (Figure 8D,E) was increased in these mice while those parameters remained unaltered in the left ventricle (Appendix A). Further echocardiographic analyses of the systolic/diastolic RV area revealed a dilatation of the RV in *Nfat5^(SMC)−/−^* mice that was accompanied by a significant drop in the RV fractional area change—a parameter that correlates with the ejection fraction (Appendix A). Interestingly, no difference in RVSP levels (Figure 8F) was observed when comparing *Nfat5^(SMC)−/−^* and *Nfat5^fl/fl^* mice after exposure to hypoxia for 21 days. SMC coverage of pulmonary arteries at this time point was comparable as well (Appendix A).

Collectively, our findings suggested that loss of *Nfat5* in paSMCs amplifies resistance of pulmonary arteries in the early phase of their hypoxia-triggered responses by promoting mitochondrial ROS production. In the long run, the exaggerated rise in afterload severely impaired right ventricular function and boosted hypertrophy and dilation of the right ventricle.

## 4. Discussion

Chronic exposure of mice to hypoxia serves as a well-characterized preclinical mouse model to elevate the resistance of pulmonary arteries that is caused by both arterial contraction and remodeling. Provided that the cardiac output remains stable, these responses increase pulmonary artery pressure and thus mimic some aspects of human pulmonary hypertension [28]. We have exploited this model to analyze the basic role of the transcription factor NFAT5 in controlling paSMC functions in the hypoxic lung by specifically ablating *Nfat5* in SMCs. While the systemic knockout of *Nfat5* in mice causes renal atrophy and early lethality [29], SMC-specific disruption of *Nfat5* in adult animals did not result in significant structural or functional phenotypic deficits including blood pressure, heart function, and structure of the vascular system as we previously described [14]. This suggests at least limited activity of NFAT5 in vascular SMCs under physiological conditions. Osmotic or biomechanical stress, however, rapidly elevates the cytoplasmic level of NFAT5 and induces its translocation into the nucleus [12,13]. This process appears to be dependent on several posttranslational modifications including the phosphorylation of NFAT5 by phospholipase Cγ1 upon exposure of cells to osmotic stress [30,31] or by the mitogen-activated (MAP) kinases JNK and c-Abl kinase in biomechanically stressed vascular SMCs [12,13]. Here, we identified hypoxia as a novel and relevant trigger for the expression, nuclear translocation, and activation of NFAT5 in vascular SMCs. While the specific context-dependent processing of NFAT5 preceding its transcriptional activity remains to be elucidated, it is likely related to the activity of JNK as it is stimulated by hypoxia in paSMCs [32]. In addition, palmitoylation of proteins—another modification of NFAT5 required to enter the nucleus [13]—was demonstrated in hypoxia-exposed pulmonary arteries [33].

Concerning its functional features, NFAT5 was originally described as an osmoprotective transcription factor [30] that appears to control the expression of different gene sets in a context-, cell-, and time-dependent manner [34]. For instance, NFAT5 regulates the expression of TNFα in osmotically stressed T-cells [35] and enhances the transcriptional function of NFκB in LPS-stimulated cells [36]. Our results, however, did not indicate that loss of *Nfat5* in SMCs affects NFκB-dependent gene expression in the hypoxic lung. Likewise, the *Nfat5*-regulated expression of cytokines as reported for mesothelial cells and macrophages [15,37] was not reflected by our transcriptome analyses. Moreover, prototypic SMC markers (e.g., *Myh11, Tagln, Acta2*) were not altered as was reported for angiotensin II-stimulated vascular SMCs [10]. In contrast, genes encoding proteins that regulate SMC contraction (e.g., *Plcb1, Gucy1b, Rock1/2, Prkg1*) showed diminished expression in hypoxia-exposed *Nfat5^(SMC)−/−^* mice. However, considering the elevated RVSP in hypoxia-exposed *Nfat5^(SMC)−/−^* mice indicating an increase in pulmonary artery resistance, this is likely to represent a secondary transcriptional response intended to dampen a pathological level of paSMC contraction rather than a cause.

Among other observations, the most intriguing result of this study is by far the massive shift in the expression of genes attributed to cellular metabolism in hypoxic lungs of *Nfat5^(SMC)−/−^* mice. Especially, genes attributed to mitochondrial respiration or OXPHOS were transcribed in lungs of *Nfat5^(SMC)−/−^* mice or *Nfat5^−/−^* mpaSMCs while being exposed to hypoxia. In line with these observations, oxygen consumption of *Nfat5^−/−^* mpaSMCs is elevated under hypoxic conditions. At the same time, the unusual accumulation of several metabolites may indicate defective or ineffective processing of substrates and/or overload of metabolic pathways, whose efficacy is limited under oxygen deprivation.

However, the mechanism by which NFAT5 directly or indirectly restricts the expression of a broad range of genes remains unclear. In fact, dependent on cell type and environmental condition, NFAT5 may adjust gene expression by at least three different modes including (i) direct stimulation of target gene transcription [38], (ii) enhancement of gene expression by forming enhanceosomes with other transcription factors [36], and (iii) suppression of gene expression by recruitment of the DNA methylase DNMT1 to the promoter of target genes [23]. Moreover, NFAT5 has been shown to epigenetically suppress the peroxisome proliferator-activated receptor gamma (PPARγ) [24], which controls several levels of nutrient and energy metabolism [39]. In fact, our transcriptome data acquired from the lungs of hypoxia-exposed *Nfat5^(SMC)−/−^* mice indicates that PPAR-dependent gene expression, as well as gene sets involved in the control of associated cellular functions (e.g., lipid/glucose metabolism), were upregulated. However, PPAR-mediated responses alone may not explain the observed effects, as treatment of hypoxia-exposed mice with the PPARγ agonist rosiglitazone attenuates rather than promotes the increase in pulmonary artery pressure and right ventricular hypertrophy [40]. While we identified several other metabolism-associated genes whose expression were up- or downregulated in hypoxic *Nfat5^−/−^* mpaSMCs (e.g., *Ucp1* and *Ldha,* data not shown) their individual functional impact could not be attributed to the observed paSMC phenotype. Likewise, elevated transcript levels of metabolism-associated genes may constitute a response to multiple metabolic deficits rather than a primary cause of the improved mitochondrial respiration.

In general, our study identifies NFAT5 as a transcriptional determinant that is required to restrict several instances of energy metabolism-associated gene expression rather than a specific metabolic feature during the adjustment of paSMCs to hypoxia. This feature of NFAT5 appears to be required to attenuate mitochondrial respiration in hypoxic *Nfat5^−/−^* mpaSMCs and to avoid amplified production of ROS. In paSMCs, mitochondrion-derived ROS are thought to influence ion channel activation, consecutively increasing the intracellular calcium level, cellular contraction, and thus hypoxic pulmonary vasoconstriction (HPV) [1,41]. Indeed mitochondrion-targeting ROS scavengers such as MitoTEMPO were reported to attenuate pulmonary artery resistance in animal models of chronic hypoxia [42,43]. In this context, our results suggest that baseline constriction, as well as acute constriction of pulmonary arteries in response to declining oxygen levels, is not altered in *Nfat5^(SMC)−/−^* versus *Nfat5^fl/fl^* mice. NFAT5 is therefore dispensable for maintaining the sensitivity or acute responsiveness of paSMCs to hypoxia, which corresponds to the almost unaltered respiration and ROS generation of *Nfat5*-deficient paSMCs observed in a normoxic environment.

However, chronic hypoxia stimulated NFAT5 activity that is indispensable for restricting pulmonary artery resistance as RVSP values increased upon loss of *Nfat5* within 7 days in *Nfat5^(SMC)−/−^* versus *Nfat5^fl/fl^* mice. We consider this mechanism dependent on mitochondrial ROS as acute administration of MitoTEMPO normalized the RVSP in hypoxia-exposed *Nfat5^(SMC)−/−^* mice. Notably, the applied concentration of MitoTEMPO effectively neutralized the increment in RVSP evoked by SMC-specific ablation of *Nfat5.* Although, in principle, augmented muscularization of small pulmonary arteries may also elevate the level of vascular resistance and RVSP values, this type of structural remodeling was not affected by loss of *Nfat5*—neither in mice exposed to hypoxia for 7 days nor 21 days. Surprisingly, RVSP values showed no difference between *Nfat5^fl/fl^* and *Nfat5^(SMC)−/−^* mice at the latter time point while the right ventricle function of *Nfat5^(SMC)−/−^* mice was severely impaired as indicated by a decrease in fractional shortening, fractional area change, and pulmonary acceleration/ejection time ratio (PAT/ET). In fact, lower PAT/ET values are usually correlated with increased RVSP values [19]. However, both the reduction in RV fractional shortening and factional area change suggest a decline in the RV ejection fraction that may influence PAT and RVSP values under these conditions. Moreover, a comparison of the RV areas after exposure to hypoxia for 21 days indicates a remarkable dilation in *Nfat5^(SMC)−/−^* mice while the RV area in *Nfat5^fl/fl^* mice shows an opposite trend. Collectively, these findings point to a severely impaired heart function that rapidly develops within the early phase of chronic hypoxia in *Nfat5^(SMC)−/−^* but not *Nfat5^fl/fl^* mice.

In conclusion, there is a growing body of evidence highlighting the relevance of adequate metabolic responses of paSMCs to hypoxia for controlling the resistance and remodeling of pulmonary arteries. In this context, our study suggests that adjustment of the paSMC OXPHOS-related metabolism is required to maintain a tolerable arterial resistance level in response to hypoxia. To this end, the hypoxic environment stimulates activation of NFAT5 to restrict metabolism-associated gene expression. This suppressive function of NFAT5 appears to be relevant during the early phase of hypoxia responses and limits mitochondrial ROS production and augmentation of pulmonary artery resistance. As such, this mechanism exerts important cardioprotective effects by preventing a detrimental rise in right ventricular afterload.

## Figures and Tables

**Figure 1 cells-10-03293-f001:**
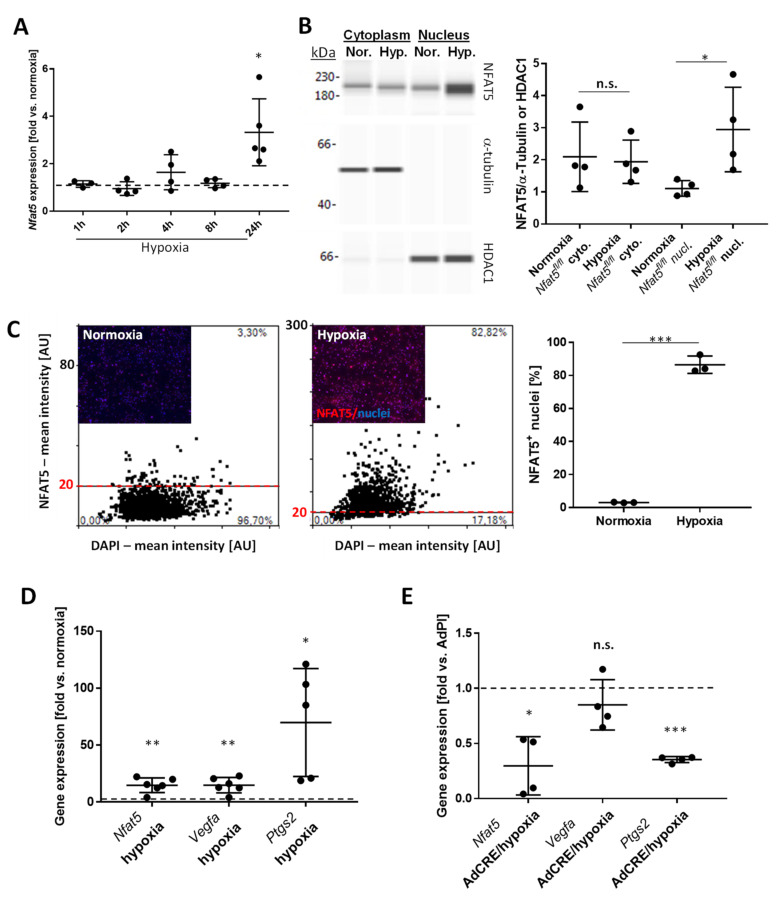
*Nfat5* expression in mouse vascular SMCs. (**A**) Murine aortic SMCs (maoSMCs) from *Nfat5^fl/fl^* mice were exposed to hypoxia for the indicated periods. The mRNA expression of *Nfat5* was analyzed by qPCR with *Actb* as reference (* *p* < 0.05 vs. normoxia (set to 1: dashed line), one-sample *t*-test, *n* = 4–5). (**B**) Capillary electrophoresis of cytoplasmic and nuclear protein fractions of maoSMCs exposed to normoxia or hypoxia (24 h) and immunodetection of NFAT5 (not significant (n.s.) vs. cytoplasm/normoxia, * *p* < 0.05 vs. nucleus/normoxia, *n* = 4). (**C**) Immunofluorescence-based detection of NFAT5 ((**C**), red fluorescence) and automated assessment (threshold level was set to 20, dashed red line) of the percentage of NFAT5-positive nuclei in maoSMCs after exposure to hypoxia for 24 h (*** *p* < 0.001 vs. normoxia, *n* = 3). (**D**,**E**) qPCR-based analysis of *Nfat5, Vegfa,* and *Ptgs2* mRNA expression in maoSMCs exposed to normoxia or hypoxia for 24 h ((**D**), * *p* < 0.05, ** *p* < 0.01 vs. normoxia (set to 1: dashed line), *n* = 5–6) as well as after genetic ablation of *Nfat5* (AdCRE, AdPl treatment served as control) and exposure to hypoxia for 24 h ((**E**, * *p* < 0.05, *** *p* < 0.001, n.s. vs. AdPl/hypoxia (set to 1: dashed line), *n* = 4).

**Figure 2 cells-10-03293-f002:**
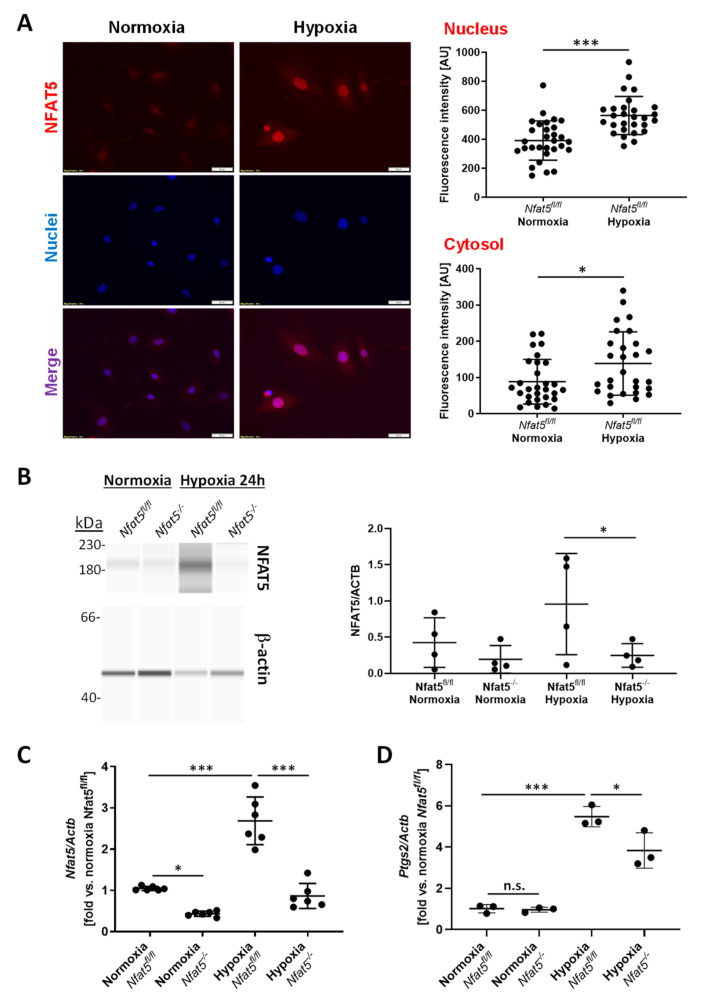
*Nfat5* expression in mouse pulmonary artery SMCs. Murine pulmonary artery SMCs (mpaSMCs) from *Nfat5^fl/fl^* mice were exposed to normoxia or hypoxia for 24 h. (**A**) Immunofluorescence-based detection of NFAT5 (red fluorescence) and assessment of cytosolic and nuclear fluorescence intensity (* *p* < 0.05, *** *p* < 0.001 vs. normoxia, the results of 19–20 randomly selected regions of interest (ROI) from one experiment are shown, scale bar: 50 µm). (**B**) Immunodetection of NFAT5 in lysates of mpaSMCs treated with AdCRE adenovirus to knockout *Nfat5* (*Nfat5^−/−^*) or AdPl adenovirus as a control (*Nfat5^fl/fl^*) after capillary electrophoresis. β-actin served as the loading reference (**p* < 0.05 as indicated, *n* = 4). (**C**,**D**) mRNA expression of *Nfat5* and *Ptgs2* was analyzed by qPCR in normoxia- and hypoxia-exposed *Nfat5^−/−^* and *Nfat5^fl/fl^* with *Actb* as reference (* *p* < 0.05, *** *p* < 0.001 as indicated, n.s.—not significant, *n* = 3–6).

**Figure 3 cells-10-03293-f003:**
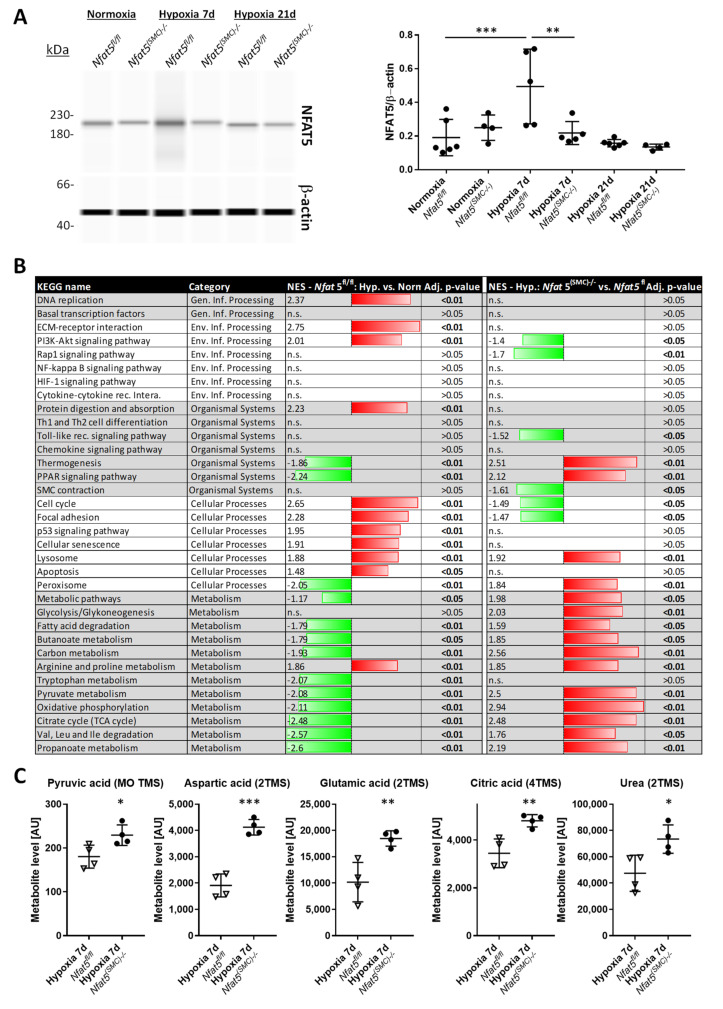
Gene set enrichment and metabolome analysis of *Nfat5^fl/fl^* and *Nfat5^(SMC)−/−^* lungs exposed to normoxia/hypoxia for 7 days. (**A**) Capillary electrophoresis and immunodetection of NFAT5 in lysates of lung lobes from *Nfat5^fl/fl^* and *Nfat5^(SMC)−/−^* mice after exposure to normoxia or hypoxia for 7 d (** *p* < 0.01, *** < 0.001 as indicated, *n* = 4–5, b-actin served as the loading reference). (**B**) RNA was isolated from lung lobes of *Nfat5^fl/fl^* and *Nfat5^(SMC)−/−^* mice exposed to normoxia/hypoxia for 7 d and analyzed by applying a whole genome microarray and gene set enrichment analysis (GSEA) according to the Kyoto Encyclopedia of Genes and Genomes (KEGG) database. The left panel lists significantly regulated gene sets (downregulated: green, upregulated: red) when comparing the normalized enrichment scores (NES) of the hypoxia (7 d)/*Nfat5^fl/fl^* and the corresponding normoxia/*Nfat5^fl/fl^* group (left panel, *n* = 4, adj *p*-value—adjusted *p*-value, n.s.—not significant). The right panel shows the comparison of hypoxia-exposed *Nfat5^(SMC)−/−^* and *Nfat5^fl/fl^* mice (*n* = 3). (**C**) Significantly regulated metabolites in lysates of lungs from hypoxia-exposed (7 d) *Nfat5^(SMC)−/−^* and *Nfat5^fl/fl^* mice were detected by gas chromatography/mass spectrometry (* *p* < 0.05, ** *p* < 0.01, *** *p* < 0.001 vs. *Nfat5^fl/fl^*, *n* = 4).

**Figure 4 cells-10-03293-f004:**
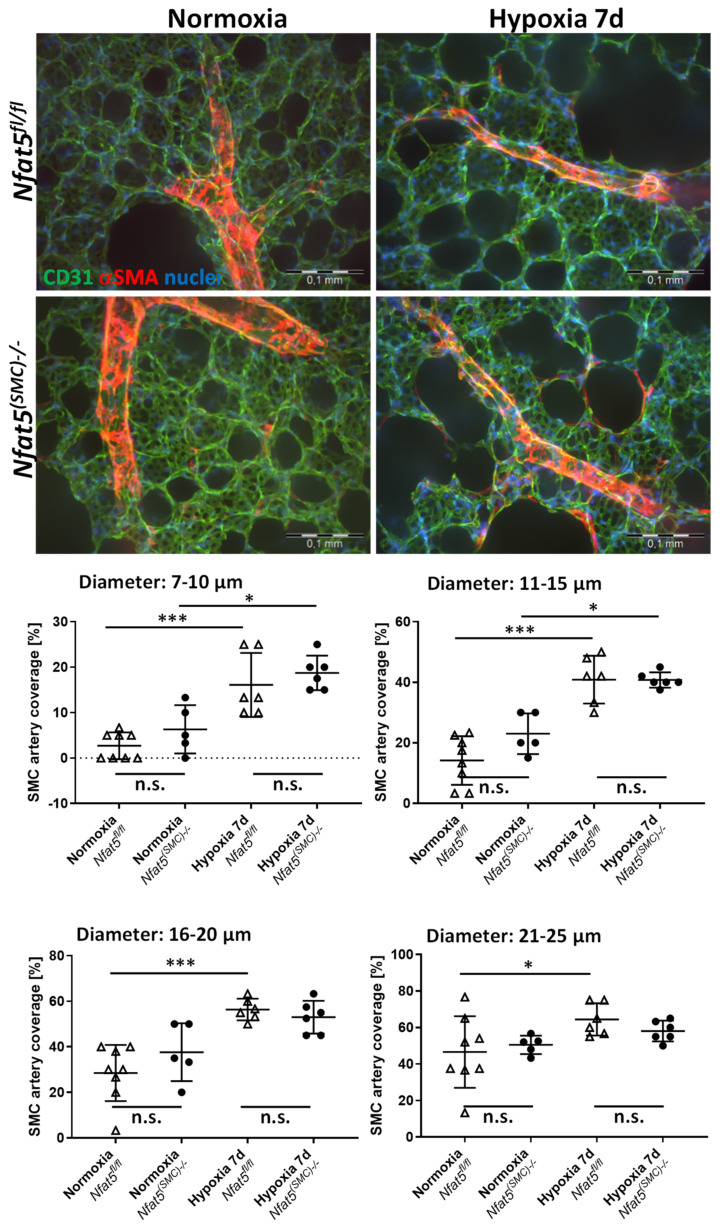
Immunofluorescence-based analysis of SMC coverage of pulmonary arteries. Vibratome sections of lungs from hypoxia/normoxia-exposed *Nfat5^(SMC)−/−^* and *Nfat5^fl/fl^* mice were processed to detect CD31 (green) and αSMA (red) by immunofluorescence-based techniques (scale bars: 100 µm). Image stacks of confocal images were morphometrically evaluated to determine the coverage of arterial segments with different caliber comprising the following groups: 7–10 µm, 11–15 µm, 16–20 µm, and 21–25 µm (n.s.—not significant, * *p* < 0.05, *** *p* < 0.001 as indicated, *n* = 5–10).

**Figure 5 cells-10-03293-f005:**
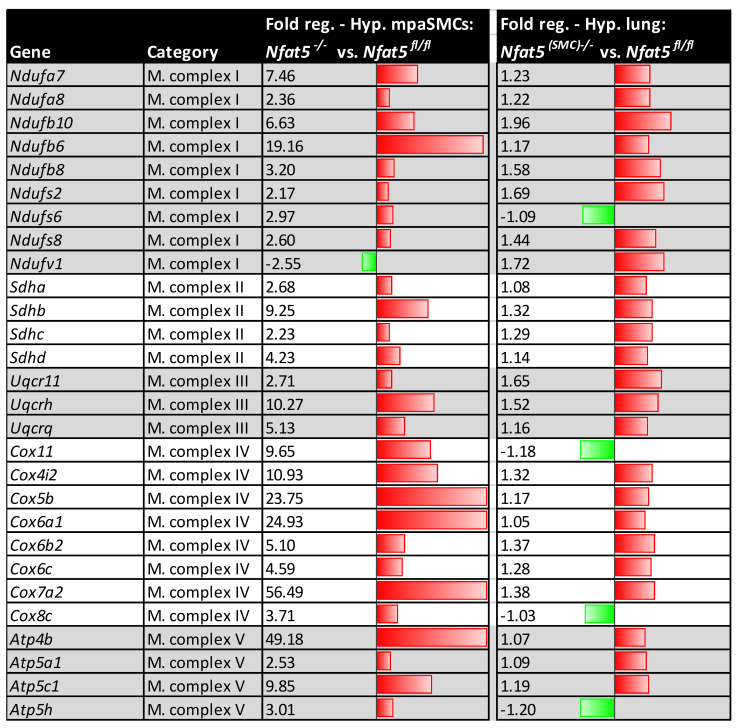
Analysis of the OXPHOS-dependent gene expression. Results of a qPCR screening (RT^2^ profiler PCR array, Mitochondrial Energy Metabolism PAMM-008ZR, Qiagen) of mpaSMCs, which were treated with adenoviral vectors (AdCRE) to knockout *Nfat5* (*Nfat5^−/−^)* or AdPl vectors as control (*Nfat5^fl/fl^*) and exposed to hypoxia for 24 h (left panel). The expression regulation of corresponding genes in hypoxic lungs (7 d) of *Nfat5^fl/fl^* and *Nfat5^(SMC)−/−^* mice is shown in the right panel (FR—fold regulation, at least two-fold regulated genes are shown in the left panel, upregulated: red, downregulated: green).

**Figure 6 cells-10-03293-f006:**
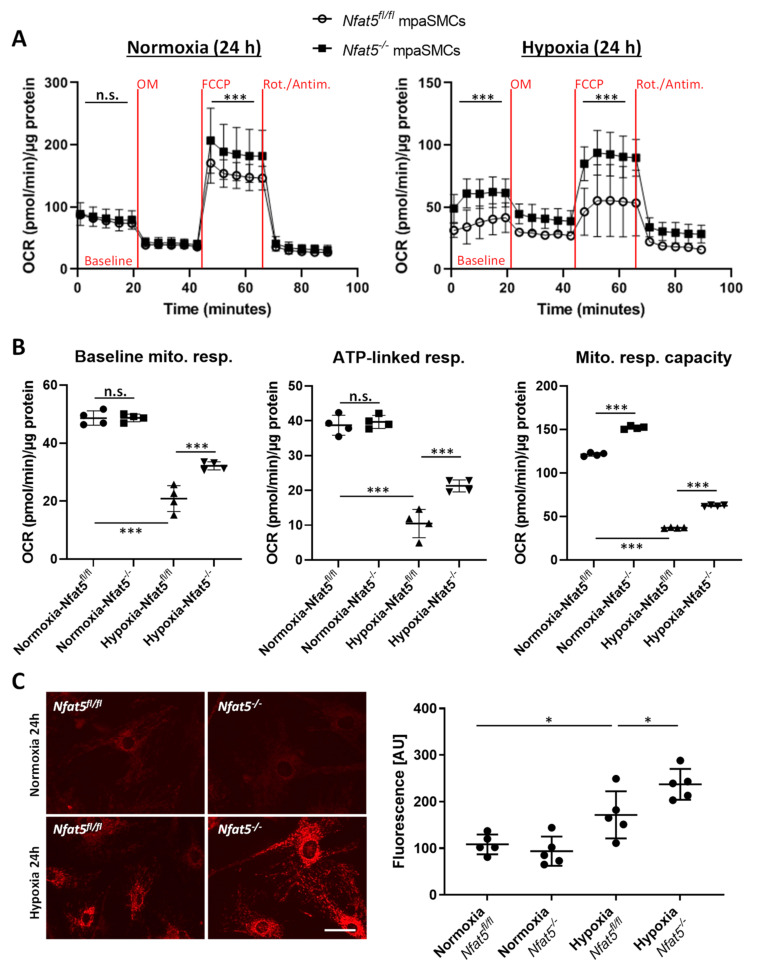
Functional analysis of hypoxia-exposed mpaSMCs. Mitochondrial respiration was assessed by analysis of the oxygen consumption rate (OCR) using a Seahorse XF analyzer. mpaSMCs were treated with an adeno-associated virus to overexpress CRE recombinase (*Nfat5^−/−^*) or GFP (*Nfat5^fl/fl^*) and exposed to normoxia/hypoxia for 24 h prior to the analysis. (**A**) OCR was assessed while treating the cells with oligomycin (OM, complex V inhibitor), FCCP (protonophore), and rotenone/antimycin A (Rot/Antim., complex I/III inhibitors) and normalized to the total cellular protein per sample (*** *p* < 0.001 vs. *Nfat5^fl/fl^*, 2-way ANOVA Šídák’s multiple comparisons test, three replicates were performed for each sample, note that some error bars were smaller than symbols, see Appendix A for further results. (**B**) Results for baseline respiration, ATP-linked respiration, and maximal mitochondrial respiratory capacity (*** *p* < 0.001 as indicated). (**C**) Mitochondrial ROS production was assessed by recording the fluorescence intensity of a ROS-sensitive mitochondrion-selective probe (red) in *Nfat5*-deficient (*Nfat5^−/−^)* or control (*Nfat5^fl/fl^*) mpaSMCs, exposed to hypoxia for 24 h (* *p* < 0.05 as indicated, one out of three experiments performed in pentaplicate, scale bar: 50 µm).

**Figure 7 cells-10-03293-f007:**
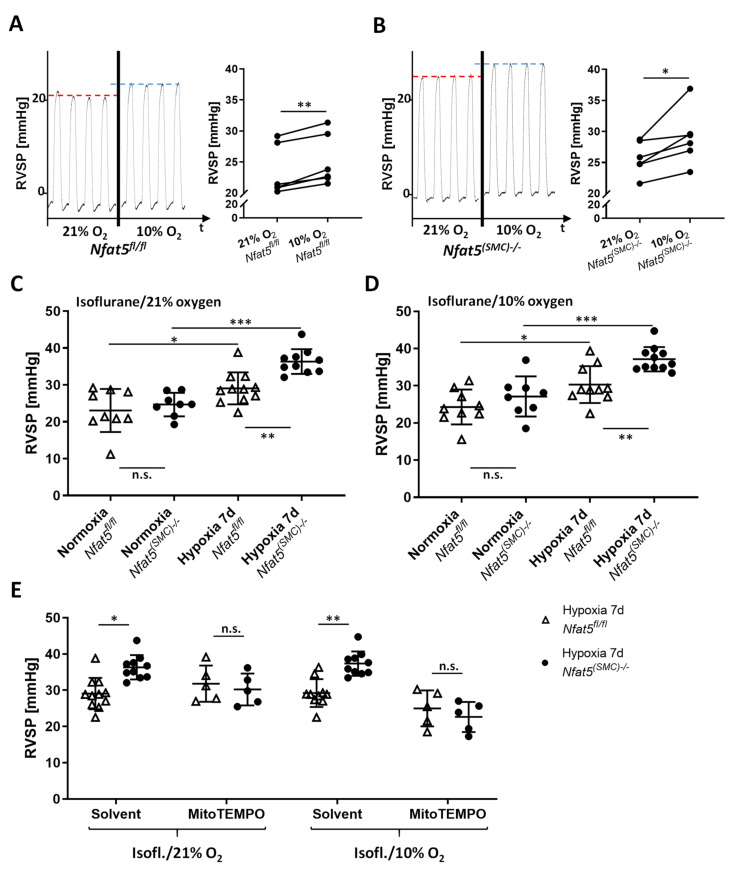
Right ventricular systolic pressure (RVSP) in *Nfat5^fl/fl^* and *Nfat5^(SMC)−/−^* mice. (**A**,**B**) The right ventricular systolic pressure (RVSP) was determined in anesthetized mice and was increased while decreasing the oxygen concentration of the isoflurane/oxygen mixture (recording of normoxia-exposed *Nfat5^fl/fl^* and *Nfat5^(SMC)−/−^* mice, respectively, * *p* < 0.05, ** *p* < 0.001 as indicated, *n* = 5). (**C**,**D**) RVSP values of normoxia/hypoxia (7 d)-exposed *Nfat5^fl/fl^* and *Nfat5^(SMC)−/−^* mice anesthetized with isoflurane/21% oxygen and isoflurane/10% oxygen (* *p* < 0.05, ** *p* < 0.01, *** *p* < 0.001 as indicated, *n* = 6–13). (**E**) RVSP was determined in anesthetized (isoflurane/21% or isoflurane/10% oxygen) mice after their exposure to hypoxia for 7 d right after treatment with the mitochondrial ROS scavenger MitoTEMPO (* *p* < 0.05, ** *p* < 0.01, not significant (n.s.) as indicated, *n* = 5–10).

**Figure 8 cells-10-03293-f008:**
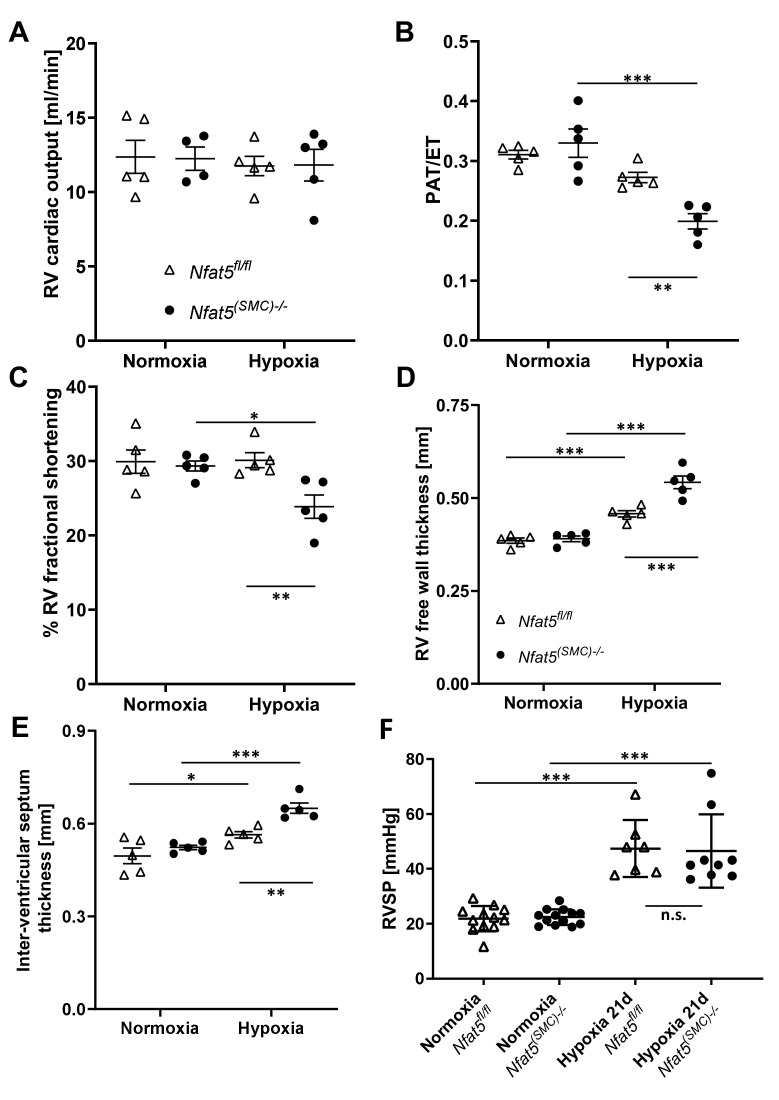
Echocardiographic analysis of functional and structural heart parameters. (**A**–**C**) Echocardiographic analysis of (**A**) right ventricular (RV) cardiac output, (**B**) pulmonary acceleration/ejection time ratio (PAT/ET, correlates inversely with the pulmonary artery resistance), and (**C**) RV systolic fractional shortening (indicates the right ventricular function) of *Nfat5^fl/fl^* and *Nfat5^(SMC)−/−^* mice exposed to normoxia or hypoxia for 21 d (* *p* < 0.05, ** *p* < 0.01, *** *p* < 0.001 as indicated, *n* = 4–5). Data are shown as mean ± SEM. (**D**,**E**) Echocardiographic analysis of structural parameters of the right ventricle (RV) of *Nfat5^fl/fl^* and *Nfat5^(SMC)−/−^* mice exposed to normoxia or hypoxia for 21 d (* *p* < 0.05, ** *p* < 0.01, *** *p* < 0.001 as indicated, *n* = 5). (**F**) RVSP was determined in anesthetized *Nfat5^fl/fl^* and *Nfat5^(SMC)−/−^* mice after exposure to normoxia/hypoxia for 21 days (*** *p* < 0.001, not significant (n.s.) as indicated, *n* = 7–13).

## Data Availability

The whole genome microarray data has been submitted to the Gene Expression Omnibus (GEO) repository. The records are available under GEO accession number GSE178468 at https://www.ncbi.nlm.nih.gov/geo/query/acc.cgi?acc=GSE178468 (accessed on 15 November 2021).

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
