# Peer review of "NFAT5/TonEBP Limits Pulmonary Vascular Resistance in the Hypoxic Lung by Controlling Mitochondrial Reactive Oxygen Species Generation in Arterial Smooth Muscle Cells"

_cells, 2021, doi:10.3390/cells10123293_

Round 1

Reviewer 1 Report

In this MS, authors Laban et al have tested the hypothesis that an alteration of vascular tone in the context of specific loss of smooth muscle NFAT5 is often associated with pulmonary vascular remodeling in driving disease etiology of PAH. To investigate deeper into mechanism, authors have utilized specific cell type knockout hypoxia models invitro and invivo and used a wide array of molecular characterization tools such gene expression, Immuno fluorescence, protein expression, Gene array, and assessment of mitochondrial capacity. Authors have attempted to distinguish the role of smooth specific role of NFAT5 in the context of HPV – a concordant and convincing case has not been established – the major drawback associated with this study, suggesting NFAT5 could not play a role in HPV in a sustained manner, just an acute effect. Here are the comments –

  1. Authors are recommended to do a minor spell check
  2. What is the rationale for choosing 500 MOI for the viral transduction of Nfat5? Any preliminary findings? How does that translate invivo?
  3. Authors have chosen to use mouse aorta or pulmonary smooth muscle cells preferentially to address selective question – a major concern. Authors are recommended to produce the entire data set based on the pulmonary SMC and replace the aorta SMC data. An outcome from aorta SMC just represents as a model rather than its translation into HPV – especially Fig 1
  4. A conventional hypoxia animal model demonstrates a high pressure on Day 21 (as shown in Fig 6) although the vasoconstriction starts at Day 7, which is subtle enough to translate into a RVSP – authors should acknowledge this. In this context, though NFAT5 is elevated in Wk1, it comes back to basal in Wk3, suggesting that NFAT5 might not play a role in HPV in a sustained fashion – a major setback! This part needs more discussion. How does the elevation of NFAT5 on wk1 translates into HPV? This outcome vaguely implies that NFAT5 activation is not a factor which drives HPV.
  5. Similarly, the influence of NFAT5 on Hypoxic-SMC cells do not make a major impact since the seahorse outcome is not reversed completely, although statistically significant in a subtle fashion suggests that the role of NFAT5 in the mitochondrial activity of SMC under hypoxia is partial. Needs more discussion.
  6. Authors could possibly show the doppler image for PAT/ET in the supplements
  7. Fig 6 - A head-to-head comparison suggest that KO of NFAT5 in SMC lacks a complete influence on RVSP on Day 21 – a major setback for the claims made by authors in the entire story line.

Author Response

Please find our responses attached.

Reviewer 2 Report

Dear editors,

Laban and colleagues present an excellent study that links NFAT5 expression and nuclear translocation to hypoxia. This finding alone is spectacular, as is the rest of the manuscript!

In brief, Laban and colleagues show NFAT5 (TonEBP) to be increased in response to hypoxia in cultured pulmonary smooth muscle cells and investigate this in a mouse model that has cell-specific TonEBP deletion in smooth muscle cells.

The manuscript is well-written and an absolute joy to read. NFAT5 is an immensely interesting molecule that has essential functions (as stated by the authors) in cellular responses to osmotic stress (e.g. high-salt environment and the immune-system).

I only have some minor comments that I think will make the manuscript more appealing and easier to understand.

Intro:

Line 59: There seems to be a distinction between acute and chronic effects regarding HPV; For someone who is not familiar with the animal model used by the authors: Would it be classified as an acute or rather chronic model or somewhere in-between?

Methods:

If I’m not mistaken the methods for figure 4 (Seahorse) are somehow missing. The supplement has more data on the experiments performed on the seahorse platform but I can’t find any methods there, either. On the other hand, there is a method section about the images acquired using a vibratome and those results are only shown in the supplement. I think it would be a good idea to explain what the seahorse is and what the compounds used (oligomycin etc.) do in the methods section.

Results:

3.1.:

So if I understand correctly, Ptgs2 and Vegfa are both downstream targets of TonEBP. Now hypoxia leads to an increase in expression of both in maoSMC (Fig 1D) but the knockout only prevents the increase of Ptgs2 (Figure 1E). Does that mean vegfa has nothing to do with TonEBP? In line 324 Vegfa is referred to as “prototypic”, maybe the authors should clarify.

The text in line 327 refers to “Figure 1E and F”. The latter I couldn’t find.

3.2.:

The vibratome sections look amazing. It’s a real shame to put them in the supplement.

So after 21 days of hypoxia, NFAT5 levels normalize again (Figure 2A left and right panels)? Could the authors elaborate on this finding a bit more?

Figure 3B:

I’m really confused by the right side of figure 3: while on the left side the increase or decrease of a given gene (e.g. Ndufa7) is given as just fold regulation, for some reason the right side is logarithmic (to the base 2). While mathematically I think this is a-okay (if I’m not mistaken that leaves us with a fold increase of 1.23 for Ndufa7), I just can’t get over the fact that there is a number <1 next to a red bad. Would there be any way to make this less complicated by leaving out the binary logarithm?

3.4.:

I think it would be good to have some more context on mitochondrial respiration. Why did the authors choose the compounds they used? Is this some published protocol (I think they are called SUIT-protocols)?

Figure 4A right panel: after giving the oligomycin and later the rotenone the Nfat5/fl/fl group has no error bars (or the error was really small).

3.5:

For someone who would not be aware of a physiological RVSP in mice: is there some sort of range? The Nfat5 fl/fl mice seem to start at 21mmHg with an increase to 22mmHg and the SMC Nfat5 KO mice seem to start at 23 and increase to 25. Maybe the authors could briefly state that this is physiological.

I have to come back to the model used: So the animals are getting the hypoxia-treatment for 7 days and then at the end they get anaesthetized with isoflurane and 21% oxygen Fig5C and then the same (?) animals get their oxygen reduced (Fig 5D). In the methods section it is stated that this is done to rule assess the effect of oxygen on the RVSP. Now if I’m not mistaken, there is no significant difference between 5C and 5D. Could the authors elaborate on whether that is expected/a good thing?

I find it astounding that MitoTEMPO would ameliorate the pulmonary hypertension observed in the SMC TonEBP KO mice. I think the authors should elaborate a bit more on this finding (acute administration of the compound improves pulmonary hypertension that has developed over a week).

Typo:

line 95: corbon dioxide

Author Response

Please find our responses attached

Round 2

Reviewer 1 Report

No suggestions